# Investigation of Unsafe Acts Influence Law Based on System Dynamics Simulation: Thoughts on Behavior Mechanism and Safety Control

**DOI:** 10.3390/ijerph20064733

**Published:** 2023-03-08

**Authors:** Xuecai Xie, Jun Hu, Gui Fu, Xueming Shu, Yali Wu, Lida Huang, Shifei Shen

**Affiliations:** 1Department of Engineering Physics, Tsinghua University, Beijing 100084, China; 2Institute of Public Safety Research, Tsinghua University, Beijing 100084, China; 3School of Emergency Management and Safety Engineering, China University of Mining and Technology (Beijing), Beijing 100083, China; 4School of National Safety and Emergency Management, Beijing Normal University at Zhuhai, Zhuhai 519087, China; 5Academy of Disaster Reduction and Emergency Management, Ministry of Emergency Management & Ministry of Education, Beijing Normal University, Beijing 100875, China

**Keywords:** unsafe acts influence law, 24Model, behavioral mechanisms, safety control

## Abstract

In modern safety management, it is very important to study the influence of the whole safety system on unsafe acts in order to prevent accidents. However, theoretical research in this area is sparse. In order to obtain the influence law of various factors in the safety system on unsafe acts, this paper used system dynamics simulation to carry out theoretical research. First, based on a summary of the causes of the coal and gas outburst accidents, a dynamic simulation model for unsafe acts was established. Second, the system dynamics model is applied to investigate the influence of various safety system factors on unsafe acts. Third, the mechanism and the control measures of unsafe acts in the enterprise safety system are studied. This study’s main result and conclusions are as follows: (1) In the new coalmines, the influence of the safety culture, safety management system, and safety ability on the safety acts were similar. The order of influence on the safety acts in production coalmines is as follows: safety management system > safety ability > safety culture. The difference is most evident in months ten to eighteen. The higher the safety level and safety construction standard of the company, the greater the difference. (2) In the construction of the safety culture, the order of influence was as follows: safety measure elements > safety responsibility elements = safety discipline elements > safety concept elements. It shows the difference in influence from the 6th month and attains its maximum value from the 12th month to the 14th month. (3) In the construction of the safety management system, the degree of influence in new coalmines was as follows: safety policy > safety management organization structure > safety management procedures. Among them, especially in the first 18 months, the impact of the safety policy was most apparent. However, in the production mine, the degree of influence was as follows: safety management organization structure > safety management procedures > safety policy, but the difference is very small. (4) The degree of influence on the construct of safety ability was as follows: safety knowledge > safety psychology = safety habits > safety awareness, but the difference on the impact was small.

## 1. Introduction

In the safety community, unsafe behavior is generally considered to be one of the direct causes of accidents [1,2]. According to some researchers [3,4,5], about 80% of direct causes of accidents are attributable to human errors. Therefore, research on the control of unsafe behavior is essential for the prevention of accidents.

Unsafe acts, unsafe behavior, and human errors are similar concepts all referring to the human causes of accidents. The literature review seeks to investigate and represent all three aspects using unsafe acts. Currently, research in this area is primarily focused on six aspects: (1) Analyzing the existing unsafe acts, unsafe behaviors, and human errors arising from the safety management process or accidents [6,7,8]. More detailed classifications and in-depth research on the obtained results have also been conducted by some scholars [9,10]. (2) Research into the causes of unsafe acts. For example, using the accident causation model [11], accident cause correlation [12], work stress [13], safety performance [14], and individual reasons [15] to investigate the mechanism of unsafe acts. (3) Conducting a risk assessment and prediction of unsafe acts. Mathematical modeling methods are used to investigate the risk of unsafe behaviors and future developmental trends [16,17,18]. (4) Safety training. Unsafe acts obtained from the search are used to perform safety training research [19,20]. (5) Fix unsafe acts. in the case of identified unsafe acts, behavior correction is carried out using methods such as BBS (Behavior Based Safety) [21], thereby reducing the occurrence of unsafe behaviors. (6) Standardization of construction safety operations. Standard operating procedures for job safety are established in order to cultivate good safety habits [22,23]. Research on these six aspects is critical to understanding and improving unsafe acts. It can be observed that much of the existing research on unsafe acts lies at the micro (individual level) and meso (team level) levels. At the micro-level, consideration should be given to the impact of individual safety abilities on unsafe acts, such as the causes of individual unsafe acts in accidents [24,25]. At the meso-level, the group safety behaviors of a group of people or a team should be taken into account [26], such as the group effects in an emergency evacuation [27].

Modern safety management requires not only consideration of micro- and meso-level human factors. In general, the safety management systems employed in companies are extremely complex. For complex engineering systems, such as process systems, managers should consider the impact of human factors on the entire safety system [28,29], that is, safety control at the macro level. The application of human factors in complex engineering systems mainly includes the following aspects: (1) Risk assessment of human error in complex engineering systems. It primarily uses qualitative or quantitative mathematical methods to assess the probability or possibility of human error [30]. These include the oil and gas industry [31], the process operation [32], and the maintenance of marine systems [33]. (2) Research into the method of human reliability assessment. This primarily advances some human reliability assessment methods [34] that belong to methodology research. (3) Considering human factors in the system or accidents and its management, such as coalmine accidents [3,7] and the management of human factors issues [35]. This work has played an important role in the assessment and management of human factors at the level of enterprise safety systems, especially in complex system engineering.

As can be seen from the above review of research, current research on unsafe acts (human factors or human errors) is predominantly at the micro and meso levels. Most macro-level research (enterprise safety system level) is concerned with the assessment of human factor risk (reliability assessment) or the analysis of human factors in accidents. In the field of complex systems engineering, studying the management and control laws of the enterprise safety system level against unsafe acts can provide important assistance in formulating safety management strategies and human factors safety assessment, but research in this area is lacking.

Beginning with an enterprise’s entire safety management system, modern safety management should consider not only the causes of unsafe acts at the individual level, but also the control and enforcement of unsafe acts at the organizational level. At the same time, the positive and negative feedback phenomena of the overall safety system must also be considered [35,36]. Therefore, in the study of control laws for factors affecting unsafe acts within the enterprise safety management system, the safety system is best used as part of an accident causation model for general research such as AcciMap, 24Model, HFACS, STAMP [37,38], and so on. In the current study, 24Model, which is based on a comprehensive and comparative study of accident causal models [1] and is more in line with modern safety management, has been used to investigate the internal causes of safety management system control laws governing unsafe acts. In theory, this information can be used to formulate methods or control measures for unsafe acts at the aggregate level. 

Since there are a variety of factors that affect the safety management system, it is extremely difficult to verify the influence of a certain factor in the safety management system on unsafe acts during the course of daily safety management. Leveson [39] used system dynamics modeling technology to investigate the dynamic control of various accident causes in safety management using the STAMP model. Xu and Luo [40] used a system dynamics model to conduct a simulation study on a risk control strategy for the unsafe behavior of air traffic controllers. This study draws on these methods, using 24Model as the theoretical model for safety management and system dynamics modeling to conduct simulation research into the control of unsafe acts within the safety management system. In this study, the research team used 24Model to analyze the causes of 84 coal and gas outburst accidents in China from 2008 to 2018 [7] and formed a comprehensive pathway of accident causes and their probability of occurrence [12]. Based on this data, the analysis began at the macro level of the enterprise safety management system and was used to investigate the simulation law of managing and controlling unsafe acts.

This study takes the coal and gas outburst accidents in China as the research object and uses system dynamics modeling to investigate the control law of the unsafe acts caused by the internal factors of the enterprise safety management system. This study focuses on the influence of safety ability on unsafe acts and various organizational factors (safety culture, safety management system). Simultaneously, the influence of various factors such as safety culture, safety management system, and individual safety ability on their safety level was also studied. Finally, the mechanism of unsafe acts (human error) in the enterprise safety system is analyzed, and the control measures for unsafe acts (human error) are proposed from three aspects of safety culture, safety management system, and individual safety ability. This study thus provides a theoretical reference for modern enterprise safety management, in particular the management and control of unsafe acts.

## 2. Data Sources

### 2.1. Accident Causation Model

It is necessary to determine the causes of unsafe acts in the system to study their management and control laws. An accident causation model, which resolves the cause of an accident and its impact relationship [1,41,42,43], can be used to provide a clear explanation of the causal relationship between unsafe acts [44,45]. The 24Model (cf. Figure 1), which is more in line with modern safety management, was selected for use in this study on the basis of a comparative study of various accident models used to investigate the law of unsafe dynamic management.

The 24Model has two structural forms: static and dynamic. In the static 24Model, the causes of accidents at the different levels and stages and the relationship between the causes can be seen. The dynamic 24Model shows the safety system control of the enterprise and the feedback process. Therefore, the static 24Model was used to analyze the causes of accidents, and the dynamic 24Model was used to guide enterprise safety management. Individual unsafe acts in the 24Model are influenced by the safety culture, safety management system, safety ability, and safety conditions. In terms of behavior control, the control process for individual unsafe acts is mainly affected by the “safety culture–safety management system–safety ability” correlation because the safety conditions are innate. Accordingly, this study investigates the influence of individual unsafe acts on the management and control, primarily to investigate the mutual relationship between the four elements of the safety management system as a result of the “safety culture–safety management system–safety ability–safety acts” condition.

**Figure 1 ijerph-20-04733-f001:**
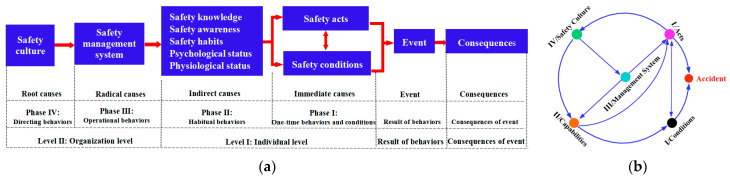
The two structural forms of the 24Model. (**a**) Static 24Model [1] and (**b**) dynamic 24Model [12].

### 2.2. Unsafe Acts Causality Analysis in Coal and Gas Outburst Accidents

The 24Model considered safety management work as a process of behavioral control. In the present study, the research team used the static 24Model to analyze 84 coal and gas outburst accidents [7] and formed a complete probability of occurrence path for accident causes, influences, and accidents of “safety culture–safety management system–safety ability–safety acts” [12]. By summarizing the cause of the accident according to the logical relationship between the accident cause modules of the 24Model, the mutual influence relationship between the four “safety culture–safety management system–safety ability–safety acts” within the enterprise was obtained (cf. Figure 2).

## 3. System Dynamics Model for Unsafe Acts

### 3.1. System Dynamics Model

In coalmines, the control and coupling relationship between unsafe acts is a complex system control process that is affected by a large number of factors. For this reason, it is necessary to choose software that is consistent with the attributes of systems science and behavioral science for simulation research.

The system dynamics model was proposed by Forrester [46] of the Massachusetts Institute of Technology. The main idea behind this model is the interactive relationship between the internal factors of the system and the software that simulates the internal response relationship of the system according to the mechanism of positive and negative feedback.

### 3.2. System Dynamics Modeling

Consistent with the causal relationship among causes shown in Figure 2, a dynamic model for the unsafe acts in the system used during the coal and gas outburst accident was constructed using state variables, rate variables, and ancillary variables. In the modeling process, the actual inter-influence relationship between the factors within and between levels was considered. As a result, a delayed impact relationship was established between some levels, such as the “safety culture–safety management system”. Delayed impact relationships were also established at some levels, such as the “responsibility assignment–safety management system”. Using the Vensim software, the dynamic simulation model for the unsafe acts of the system during the coal and gas outburst accidents was set up.

### 3.3. Construction of the Simulation Equation

The accuracy of the causal relationship in the enterprise safety management system will affect the final results of the simulation. In this study, 24Model was used as the theoretical analysis tool, and the system dynamics simulation was performed using the data on the causes of the 84 coal and gas outburst accidents [7]. Thus, the accuracy of the influence relationship between the causes was tested. Finally, the dynamic model for the unsafe acts management and control system was constructed based on the influence relationship of the accident causes shown in Figure 2 and the principle of the system dynamics and using the relationship among the state, rate, and auxiliary variables. The main simulation equations used in the simulation process are as follows:Initial time = 0, end time = 24, time step = 0.25, unit: month.$SC-rate-up=(1-e^{\ln^{2}\times (\frac{SC1_{IV}\times a1+SC2_{IV}\times a2+SC3_{iv}\times a3+SC3_{IV}\times a4}{80})})\times e^{-Time\times \frac{\ln^{2}}{6}}$

In the formula: *IV* = Initial value, *SC*1*_IV_* represents the initial value of *SC*1, same as below; and *a*1, *a*2, *a*3, and *a*4 are the weight coefficients of *SC*1, *SC*2, *SC*3, and *SC*4, respectively, same as below3.$SC-rate-down=DELAY1((1-e^{-UA\times \frac{\ln^{2}}{4}}),1)$4.$SM-rate-up=DELAY1((1.5\times (1-e^{-\ln^{2}\times \frac{SC}{2}})\times (1-e^{-\ln^{2}\times (\frac{SM1\times d1+SM2\times d2+SM3\times d3}{3})})\times e^{-SM\times \ln2^{}},0.25)$5.$SM-rate-down=DELAY1((1-e^{-UA\times \frac{\ln^{2}}{4}}),0.5)$6.$SM2-rate-up=(1-e^{\ln^{2}\times (\frac{SM21_{IV}\times b1+SM22_{IV}\times b2+SM23_{iv}\times b3}{80})})\times e^{-Time\times \frac{\ln^{2}}{6}}$7.$SM3-rate-up=(1-e^{\ln^{2}\times (\frac{SM31_{IV}\times c1+SM32_{IV}\times c2+SM33_{iv}\times c3+SM34_{iv}\times c4+SM35_{IV}\times c5+SM36_{IV}\times c6}{80})})\times e^{-Time\times \frac{\ln^{2}}{6}}$8.$SA-rate-up=DELAY1((1.5\times (1-e^{-\ln^{2}\times \frac{SM}{2}})\times (1-e^{-\ln^{2}\times (\frac{SA1\times h1+SA2\times h2+SA3\times h3+SA4\times h4}{4})})\times e^{-SM\times \ln2^{}},0.25)$9.$SA-rate-down=DELAY1((1-e^{-UA\times \frac{\ln^{2}}{4}}),0.25)$10.$SA1-rate-up=(1-e^{\ln^{2}\times (\frac{SA11_{IV}\times e1+SA12_{IV}\times e2+SA13_{iv}\times e3+SA14_{iv}\times e4+SA15_{IV}\times e5+SA16_{IV}\times e6+SA17_{IV}\times e7}{80})})\times e^{-SA1\times \ln^{2}}$11.$SA2-rate-up=(1-e^{\ln^{2}\times (\frac{SA21_{IV}\times f1+SA22_{IV}\times f2+SA23_{iv}\times f3}{80})})\times e^{-Time\times \frac{\ln^{2}}{6}}$12.$SA3-rate-up=(1-e^{\ln^{2}\times \frac{SA31_{IV}}{80})})\times e^{-Time\times \frac{\ln^{2}}{6}}$13.$SA4-rate-up=(1-e^{\ln^{2}\times (\frac{SA41_{IV}\times g1+SA42_{IV}\times g2+SA43_{iv}\times g3+SA44_{IV}\times g4}{80})})\times e^{-Time\times \frac{\ln^{2}}{6}}$14.$UA-rate-up=(1-e^{\ln^{2}\times \frac{SA_{IV}}{2})})\times e^{-UA\times \ln^{2}}$

### 3.4. Initializing the Parameter Settings

Parameter initial value settingThe auxiliary Variable (AV) ranges from 0 to 100 and represents the ability of safety construction.The horizontal Variable (HV) ranges from 0% to 100% and represents the ratio of the safety value of the current factor to the maximum value.Weight coefficient setting

The weight coefficients for each factor in this study were derived from the proportional values for the causes of the 84 coal and gas outburst accidents [7]; therefore, they are authentic. However, due to the length limitations of the manuscript, this study only provides the final ratio.

Safety culture layer. a1:a2:a3:a4 = 2:6:7:7 = 0.09:0.27:0.32:0.32.Safety management system layer. b1:b2:b3 = 21:77:44 = 0.15:0.54:0.31; c1:c2:c3:c4:c5:c6:c7= 0.08:0.08:0.21:0.17:0.12:0.17:0.17; d1:d2:d3 = 0.2:0.5:0.3.Safety ability layer. h1:h2:h3:h4 = 0.5:0.1:0.2:0.2; e1:e2:e3:e4:e5:e6:e7 = 0.7896:0.0861:0.0615:0.0232:0.0137:0.0178:0.0082; f1:f2:f3 = 0.6066:0.2049:0.1885; g1:g2:g3:g4 = 0.4085:0.2022:0.2336:0.1557.

## 4. Simulation Results and Analysis

### 4.1. Simulation Results for the New Coalmine

The newly constructed coalmines were analyzed in the initial stage of safety system construction where the safety level was zero (HV = 0). Some impact on the level of enterprise safety culture, safety management system, safety ability, and safety acts was observed when different auxiliary variables were used, that is, when different safety construction standards were adopted. In this study, the AVs were set to 60, 70, 80, and 90 (representing different levels of safety construction standards) to simulate the changing trends in the safety culture, safety management system, safety ability, and safety acts. Figure 3 shows the simulation results.

Based on the simulation results of the new coalmine, (1) the safety construction of the coal mining companies, safety culture, safety management system, and safety ability initially showed an increasing trend before slowly decreasing, while safety acts showed a continuously increasing trend over the duration of the simulation. (2) Growth in safety culture, safety management system, and safety ability indicated a small difference and peaked in months thirteen through fourteen, fifteen through sixteen, and sixteen through seventeen, respectively. (3) With respect to the different safety construction standards, there was only a small difference in the times for the safety culture, safety management system, and safety ability. (4) At the same time, the higher the standard of safety construction, the higher the peak level for each factor. Therefore, for new coalmines, it is necessary to improve the standards of safety construction to improve the safety culture level, safety management system level, safety ability level, and safety acts level within a short period of time.

The rationale for these simulation results is as follows: (1) In new coalmines, the safety culture is more easily accepted by miners than the safety management system and safety ability. (2) In constructing the new coalmine safety management system, it takes about 2–3 weeks to first build the organizational structure and formulation of safety procedures. (3) Newly recruited miners must receive no less than 72 h of safety training in accordance with the “Coalmine Safety Training Regulations”, and, simultaneously, they must practice for four months under the leadership of experienced workers. Therefore, the safety ability and safety acts levels form a relative lag. (4) Increasing the safety acts level forms a negative feedback effect on the safety culture, safety management system, and safety ability, which causes the three to decrease after a certain period of time. This study proposes that for new coalmines, after a certain construction period (approximately 12 months), the safety construction situation should be evaluated and the construction plan optimized in order to achieve the best safety construction effect.

### 4.2. Simulation Results for the Production Coalmine

Each production coalmine had its own safety management system and formed a certain degree of safety culture, safety management system, safety ability, and safety acts. Their safety construction typically focuses on three things: (1) Changes in the safety culture, safety management system, safety ability, and safety acts. (2) Changes in the safety acts under the same safety level when different safety construction capabilities are adopted. (3) Changes in the safety acts under different coalmine safety levels and similar safety construction ability. Aiming at these three problems, this paper does a simulation study.

#### 4.2.1. Simulation Laws for Different Safety Factors

Similarly, safety managers at production coalmines are also concerned with changes in the safety culture, safety management system, safety ability, and safety acts. It can be observed from Figure 4 that although the simulation results are different under different safety construction standards, they show an approximate regularity. Therefore, the AVs in this study were set to 60 and the enterprise safety level variables were HV = 25%, 50%, and 75%, respectively (assuming that the safety level of the coalmine was 25%, 50%, and 75% of the optimal state at the time), and the simulation results are shown in Figure 4.

The results of the simulation law showed that there was a large difference between the production coalmine and the new coalmine. For safety culture, the law’s trend initially showed an increase before decreasing. Enterprises with lower levels of safety (HV = 25%) experienced a significant change in safety culture, with a short peak period. Enterprises with a higher level of safety (HV = 75%) showed a relatively stable safety culture and continued to maintain a high level where the period of peak safety was long and steady. These results show that once an enterprise’s safety culture is formed it can continue to be maintained.

The safety management system and safety culture showed different development laws. In the early stages of the construction of the safety management system, the companies with a low safety level (HV = 25%) demonstrated continuous growth after a short period of decline, and after peaking they were affected by the negative feedback and started to decrease. The companies with a high safety level (HV = 75%) showed a relatively long period of reduction, which indicates that improving the safety management system in companies with a high safety level will disrupt the previously stable safety deployment (trough period). At the end of this period, they continued to grow until they reached a peak value that was much higher than that of companies with lower levels of safety construction.

There was a small similarity in the pattern of changes in the individual safety ability and safety management system, and both initially decreased before increasing. Changes were relatively small for enterprises with low levels of safety (HV = 25%). However, enterprises with higher levels of safety (HV = 75%) were significantly impacted by negative feedback, and the final peak in safety capability was the highest.

The changes in the safety acts showed different laws. The companies with a low safety level (HV = 25%) had a low level of safety acts. Therefore, if the safety construction is increased, the safety acts will show significant growth, and the effect would be evident. The companies with a higher safety level (HV = 75%) had a higher level of safety acts for their employees. Increasing the safety construction will increase the employees’ safety acts within a short period of time. On the other hand, the effect of the employees’ safety acts may be reduced by negative feedback.

The above simulation results show that (1) in an enterprise with a low safety level, when the safety construction was increased, the safety culture also increased. Due to the overall safety level of the enterprise, the safety culture could not maintain a high peak value, and the improvement in the safety management system, safety ability, and safety acts were significant. (2) When safety construction was increased for enterprises with high levels of safety, their safety culture did not exhibit significant changes but continued to maintain a high peak value of safety. Due to the breaking of the original safety balance, the safety management system, safety ability, and safety acts first decreased before increasing. This warns coalmine managers that in companies with higher safety levels, safety construction measures can be optimized and adjusted to a certain extent, but large-scale safety construction should not be carried out. This disrupts the high-quality safety management methods that miners have been accustomed to and makes miners uncomfortable with this safety system, which may lead to an overall reduction in the use of the safety management system, safety ability, and safety acts.

#### 4.2.2. Impact on the Safety Acts with Different Safety Building Ability at the Same Safety Level

A variety of safety construction standards were used to study the influence of the safety acts when the production coalmine was at the same safety level. For the purposes of this study, the simulations were conducted at a safety level of 25%, 50%, and 75% (assuming that the safety level of the coalmine is 25%, 50%, and 75% of the optimal state), and the construction was carried out based on a safety construction energy of 70, 80, and 90. Figure 5 shows the change in the safety acts.

The simulation results show that (1) the safety act initially maintains a small increase at approximately 4 months. Subsequently, the safety act first showed a transition from a decreasing trend to an increasing trend. (2) There is no evident difference in the impact of the different safety construction capabilities on the safety acts in the early stage of safety construction (approximately 4 months). (3) The final value of the safety acts for companies with lower safety levels (HV = 25%) was slightly higher than that of companies with higher safety levels (HV = 75%). (4) For different safety levels, the fluctuation trend for the safety acts was slightly different. When the safety level was low (HV = 25%), the fluctuation in the attenuation was low. The fluctuation in attenuation was greater when the level of enterprise safety was high (HV = 75%). (5) The stronger the safety construction ability, the faster the growth rate of the safety act, and the smaller the fluctuation of its attenuation.

The reasons for the above situation are analyzed as follows: (1) Increasing the safety construction of the enterprise and safety acts increased the influence of the safety culture and safety management system. Subsequently, the negative feedback resulted in a decreasing trend and the safety acts resulted in a steady growth with a stable formation of safety acts. (2) During the initial stages of safety construction (approximately 4 months), the different safety construction abilities did not affect the safety acts, the main reason being that the safety acts of miners were not stable. Therefore, the main factors affecting the safety acts are the binding forces of safety culture and safety management system. After a stable safety act was formed in the latter stages, the influence of the different safety construction abilities on the safety act presented significant differences. (3) The higher the safety construction ability, the stronger the adjustment ability of the coal mining enterprise itself. Therefore, the safety act is less affected by the feedback effect and there is less fluctuation.

#### 4.2.3. Impact on the Safety Acts of Different Safety Levels and the Same Safety Building Ability

Enterprise managers are also concerned about changes under different levels of safety and the same safety ability. To investigate this problem, this study performed the following three sets of scenario simulations to study this problem. The changes in the safety acts under three different safety levels (HV = 25%, 50%, and 75%) were studied under a safety construction ability of 70, 80, and 90, respectively (AV = 70, 80, 90). Figure 6 shows the simulation results.

Simulation results have shown that (1) production coalmines already have a certain level of safety. Therefore, the construction of the safety ability increased, and, within a certain period of time, the level of safety acts also increased. Once it has increased to the maximum value, the negative feedback of the safety act affects the safety culture, safety management system, and safety ability, which in turn leads to a decrease in the level of safety acts. After reaching the bottom, it is again affected by negative feedback, and the changes in the safety acts increase. (2) Under the same safety construction ability, the lower the safety level of the enterprise, the smaller the “negative feedback” effect, that is, the smaller the fluctuation trend. Companies with a higher level of safety had a larger trend in volatility. (3) Building a safety system based on the different construction standards. The stronger the construction ability, the smaller the “negative feedback” effect, that is, the smaller the fluctuation trend. This is consistent with the simulation results above. (4) As the enterprise’s safety level increases, the peak value of the final safety act level increases.

The reasons for the above-mentioned phenomenon are roughly as follows: (1) The negative feedback effect caused by the change in the safety act level is often received at a time when the safety level in production coalmines is low. When the safety level is high, it is often paralyzed by the high safety state of the enterprise itself, and it is not easy to observe the reduction in the safety acts level. Therefore, for safe production, it is necessary to detect the occurrence of unsafe acts in time and make timely corrections. (2) The higher the safety construction ability of an enterprise, the stronger its own adjustment ability. Thus, the safety acts are less affected by the “feedback” effect and a lower fluctuation is presented.

### 4.3. Simulation Results at the Single Accident Cause Level

When improving the safety culture of construction, safety management system, and safety ability, the enterprise’s safety management is approximately correlated to its impact on the individual safety acts. For this reason, separate studies were conducted in this study, and the simulation results are shown in Figure 7.

#### 4.3.1. Influence of the Safety Culture on the Safety Acts

The influence of the safety culture on the safety acts is shown in Figure 7a. The simulation results showed that (1) safety culture has a significant impact on the formation of safety acts. (2) For the same safety culture level, changing the construction standards for the safety culture has a small effect on the increase in the safety acts in the early stages of construction (6 to 8 months). (3) For the continuous construction of the safety culture, a higher safety culture construction standard resulted in a faster growth rate for the safety acts and made it more sensitive to the occurrence of unsafe acts. (4) The safety culture has a certain negative feedback effect on the safety acts (the trough period of safety acts). During this trough period, the stronger the safety culture building ability of the company, the lower the trough value and the faster the rebound. (5) An interesting phenomenon is that the influence of the safety culture on the safety acts is larger in companies with lower safety levels, that is, the final peak of safety acts is higher.

The reasons for this are that (1) the safety climate of the enterprise formed by the safety culture affects the safety acts of the employees as a whole, and this effect needs to have a slow period. Therefore, changing the construction standards of the safety culture will have a lower impact on the improvement of the enterprise safety atmosphere within a short period; therefore, the impact on the safety acts will not be clearly displayed. Due to the continuous construction of a safety culture, companies have gradually formed a good safety culture and safety atmosphere, and various safety concepts have already been integrated into the practical work of miners. In this study, the influence of the safety culture on the safety acts was gradually reflected. (2) The stronger the safety culture construction ability of the enterprise, the higher the safety culture level and safety atmosphere of the enterprise; therefore, the growth of the safety acts will be performed faster. Simultaneously, the trough period for the safety act is shorter. (3) The stronger the safety culture construction ability of the enterprise, the stronger the safety culture and safety atmosphere, which forms a fixed impact on the safety act. Therefore, the impact of the construction of the continuous safety culture on the safety acts becomes weaker. These findings suggest that the content and method of the safety culture are constructed over time, so as to form a “new safety culture” that will be more helpful to the growth of safety acts.

#### 4.3.2. Influence of the Safety Management System on the Safety Acts

Figure 7b shows the influence of the safety management system on safety acts. From the simulation law, the safety management system’s simulation results for the safety acts are similar to those of the above simulation results: (1) Under the same safety management system level, changing the safety management system construction standards has a slight effect on the increase in the safety acts in the early stages of construction (4–6 months). The possible reason is that it takes a considerable amount of time to fully develop the safety management organizational structure and safety procedures in the safety management system to the point where it is fully accepted by employees and forms stable safety acts. (2) With the continuous construction of the safety management system, the higher the construction standard of the safety management system, the faster the growth rate of the safety acts, and the more sensitive the occurrence of unsafe acts. So also, is the impact of the safety culture construction on the safety acts.

#### 4.3.3. Influence of the Safety Ability on the Safety Acts

The impact of building safety ability on the safety acts is shown in Figure 7c. From the simulation law perspective, the individual safety ability simulation results were similar to the simulation results for the safety culture and safety management system. (1) Under the same safety ability level, changing the construction standard of the safety ability had a small effect on the increase in the safety acts in the early stage of construction (the first 2 months). It is evident that increasing the safety ability of individuals increases the safety acts of employees. However, it takes a considerable amount of time to increase the individual ability of employees and there are certain delays in the formation of the safety acts. (2) With the continuous improvement in the safety ability, the higher the safety ability of building standards, the faster the growth rate of the safety acts, and the more sensitive the occurrence of unsafe acts. This phenomenon was similar to the influence of the safety culture construction and safety management system construction on the safety acts.

### 4.4. Simulation Results for Different Accident Cause Levels

The influence of the safety culture, safety management system, and safety ability on the safety acts were also studied when the safety construction standards were different. Safety managers typically consider systems that have a significant impact on the safety acts, including the safety culture, safety management system, and safety ability. Therefore, this study considers the influence of the safety culture, safety management system, and safety ability on the individual safety abilities under different safety levels and safety construction standards. The simulation results are shown in Figure 8.

The simulation results showed that: (1) in new coalmines, the influence of the safety culture, safety management system, and safety ability on the safety acts is not very different. (2) In production coalmines, the influence of the three above-mentioned aspects on the safety acts is expressed as follows: safety management system > safety ability > safety culture. With the improvement in the coalmine safety level and safety construction ability, this gap has become more evident. (3) During the trough period for the safety acts, the stability of the three aspects can also be expressed from a perspective considering the volatility of the impact on the safety acts as follows: safety management system > safety ability > safety culture.

In new coalmines, various safety levels are gradually being developed. The safety culture, safety management system, and safety ability, influence each other and jointly affect the formation of the safety acts. However, modern safety management requires a more complete safety management system detailed safety management organization and procedural safety execution. Therefore, the safety management system plays a more important role in the safety acts of modern companies. As a potential manifestation of the individual safety level, the individual safety ability directly affects the formation and improvement of safety acts. Therefore, improving the construction of the individual safety ability from safety knowledge, safety awareness, safety habits, and safety psychology perspective plays an important role in the formation of the employees’ safety acts. In a modern enterprise, there exists a corporate safety culture that the employees are familiar with. Consequently, if the construction of the corporate safety culture is improved further, the improvement in employee safety acts will not improve to the same extent as the safety management system and safety ability.

### 4.5. Simulation Results for the Influence of Different Factors within the Accident Cause Hierarchy

The effects of the safety culture, safety management system, and safety ability on the safety acts were studied under different safety levels and safety construction standards. In addition, the influence of the internal factors at each safety culture, safety management system, and safety ability level on its own level is important in safety management. Therefore, corresponding research was conducted in this study.

#### 4.5.1. Safety Culture Layer

This study divides the safety culture elements into four categories: safety concept elements, safety discipline elements, safety responsibility elements, and safety measure elements [7,12]. Therefore, this study simulates the effects of adjusting the construction standards of the safety culture elements on the safety culture level during different safety culture construction periods. The scenarios were observed as follows:(1)When HV = 0% (new coalmine), the safety construction ability of four types of elements was adjusted from AV = 60 to AV = 80, where the simulation results for the safety culture level are shown in Figure 9a.(2)When HV = 25% (production coalmine), the safety construction ability of the four types of elements was respectively adjusted from AV = 60 to AV = 80. The simulation results for the safety culture level are shown in Figure 9b.

**Figure 9 ijerph-20-04733-f009:**
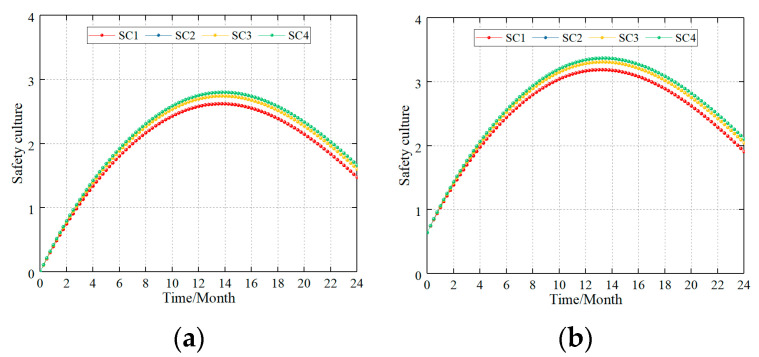
Effect of the different safety culture elements on the safety culture level. (**a**) HV = 0%, AV = 60 adjusted to AV = 80 and (**b**) HV = 25%, AV = 60 adjusted to AV = 80.

Simulation results demonstrated the following: (1) The effect of the safety culture elements on the overall safety culture initially increases before declining. (2) This is consistent with the effects of all four types of safety culture items on overall safety culture. The degree of influence on the level of safety culture was as follows: safety measures elements > safety responsibility elements = safety discipline elements > safety concept elements, and the gap was not significant.

Primarily, the reasons include the following aspects: (1) The safety culture of construction elements promotes the improvement of the overall safety culture level of the enterprise. However, after the construction reaches a certain level, it will gradually show a certain attenuation due to the influence of negative feedback. (2) There are many types of safety culture elements. Within an organization, a safety culture develops a philosophy and perception that influences the creation, growth, and maintenance of safety acts. Therefore, some safety culture elements that emphasize safety measures are closely related to the work of the employees; therefore, the improvement in the safety acts will be more prominent. The elements that emphasize safety responsibilities and safety discipline had the second highest magnitude in the effect. For the construction of the safety culture, the safety concept type is widely recognized and accepted for use, and its impact on the safety acts is not as evident as that in the first three types of safety culture elements.

#### 4.5.2. Safety Management System Layer

A complete safety management system typically consists of three parts: safety policy, safety management organizational structure, and safety management procedures [1,3,7,12]. Likewise, there was an adjustment from an AV = 60 to AV = 80 via the analogy of the safety culture layer when building the safety ability of the three elements to study the influence of the three parts on the safety management system. Figure 10 shows the simulation results.

Simulation results revealed that (1) in the new coalmine (Figure 10a), the impact of the safety management organizational structure and safety management procedures on the safety management system was delayed because the enterprise safety management procedures were not fully established or running normally. Here, the safety policy had the greatest impact on the safety management system level, and the degree of influence on the safety management system level was safety policy > safety management organizational structure > safety management procedures. (2) In the production coalmines (Figure 10b–d), the enterprise safety management system was in normal operation and had certain safety management abilities. At this point, the degree of influence on the safety management system at this time was safety management organizational structure > safety management procedures > safety policy. However, the difference in the impact of the three aspects on the safety management system level was very small and it can be considered to have the same effect. The reason for this phenomenon is that in a developed, stable and normally operating enterprise safety management system, the safety policy, safety management organizational structure, and safety management procedures interact with each other and play a collective role in the safety management system.

#### 4.5.3. Safety Ability Layer

Individual safety ability directly affects the formation and maintenance of safety acts that are primarily affected by four factors: safety knowledge, safety awareness, safety habits, and safety psychology [7]. In order to investigate their impact on the safety ability, the safety ability building standard in this study is adjusted from 60 to 80 (AV = 60 adjusted to AV = 80). Figure 11 shows the simulation data.

The simulation results showed that (1) from a data perspective, the degree of influence on the safety ability was safety knowledge > safety psychology = safety habits > safety awareness. However, the effect on the safety ability level is minimal and the approximation can be considered consistent. (2) When the individual safety ability is low, improving the construction of miners’ safety knowledge, safety psychology, safety habits, and safety awareness will improve the individual safety ability. However, when the individual safety ability is high, the construction of the miners’ safety knowledge, safety psychology, safety habits, and safety awareness will continue to be improved. Therefore, the growth rate of the individual safety ability was low. Particularly, the safety ability of an individual is reduced if affected by negative feedback.

Based on the current research, the collective influence mechanism between safety knowledge, safety psychology, safety habits, and safety awareness has not yet been investigated in safety management. However, from the simulation data, it had a similar influence and improvement. The authors of this study believe that the main cause of poor safety psychology, safety habits, and safety awareness is insufficient safety knowledge. When determining the safety knowledge, the safety psychology, safety habits, and safety awareness were improved. On the other hand, the process of improving safety psychology, safety habits, and safety awareness is similar to the process of enhancing safety knowledge.

## 5. Thoughts on Behavior Mechanism and Safety Control

The system simulation of the influence law of unsafe acts is a theoretical study of the macroscopic influence law based on a large number of accident-causes data. Simulation results in combination with the actual safety management can reflect the causes of the unsafe acts behavior mechanism and then provide countermeasure support for managing and controlling unsafe acts.

### 5.1. Enterprise Safety System

#### 5.1.1. Current Enterprise Safety System 

Safety culture, safety management system, and individual safety ability jointly influence the formation of employees’ safety acts in a company’s safety system. The following phenomena can be observed through the simulation data in Figure 8.

New coalmine. The impact of safety culture, safety management system, and individual safety ability all have a consistent impact on unsafe acts.Production coalmine. The order of influence on unsafe acts is safety management system > safety ability > safety culture. The greater the level of coalmine safety or safety construction ability, the more obvious this shortcoming.

#### 5.1.2. Unsafe Acts Occurrence Mechanism in Enterprise Safety System

Based on the simulation results and the actual safety management, this paper infers the behavioral mechanism that affects unsafe acts at the enterprise safety system level, as shown in Figure 12.

New coalmine. The enterprise safety construction has just begun, and a mature and stable safety system has not yet been formed. At this time, the corporate safety culture has not been formed. In the safety management system, responsibility assignments are unclear, personnel allocation is uneven, and safety management procedures are missing. These reasons result in insufficient individual safety knowledge, weak safety awareness, failure to develop safety habits, and failure to form safety psychology. On the whole, in the new coalmine safety system, the content of the safety culture and safety management system is only a form of a signal transmitted to employees, so there is no difference in their impact on unsafe acts. At this time, the unsound safety system causes the individual’s safety ability to be weak, resulting in the individual failing to form stable safety acts.Production coalmine. In modern enterprise safety management, a healthy enterprise pays attention to a sound safety management system and a programmed safety management process. At the same time, the operation of the safety management system directly affects the formation of individual safety abilities. Therefore, the control effect of the mandatory and binding force of the safety management system on unsafe acts will be very obvious. At the same time, employees already have certain safety abilities and are familiar with the operation of the safety management system. The level of individual safety ability will directly affect the occurrence of unsafe acts. As a safety concept, safety culture does not have the coercive and binding force of safety management, and the control over unsafe acts will become very weak at this time.

These laws tell managers that in a production enterprise, the most important weapon for controlling employees’ unsafe acts is a safety management system, including the correct setting of mechanisms, clear assignment of responsibility, sufficient allocation of safety management personnel, and a clear procedural safety management process. Strengthening employees’ safety ability (safety knowledge, safety awareness, and safety psychology) and then developing good safety habits is the most direct means to improve the occurrence of incomplete acts.

### 5.2. Safety Culture

Safety culture refers to the source and specific content of the official guiding ideology of safety at work, which can also be called the safety concept, and it should be embodied by all the individual members within an organization [1,3,7,12]. Consequently, the safety culture is not mandatory and binding, but it is the embodiment of the enterprise’s safety level.

The influence of safety culture on unsafe acts is shown in Figure 10. Based on these laws, the design of safety culture construction in this paper is shown in Figure 13.

#### 5.2.1. Current Safety Culture 

As we all know, safety culture has a certain influence on the control of unsafe acts, but it is not as obvious as the safety management system and individual safety ability. The primary reason is that safety culture lacks binding and coercive force to control unsafe acts, followed by lack of professionalism and guidance. At present, the safety culture construction of Chinese companies is relatively weak. From the status quo, it is more like a popular science culture. This paper believes that to improve the influence of safety culture, the following two aspects need to be carried out.

The constraining force of safety culture. Based on the simulation data, the ranking of the influence of safety culture elements on unsafe acts is as follows: safety measures elements > safety responsibility elements = safety discipline elements > safety concept elements. This shows that elements such as safety measures, safety responsibilities, safety disciplines, etc., are more binding for unsafe acts. This paper suggests that in the construction of safety culture, on the basis of the propaganda of safety concepts, the elements of safety culture should be diversified and specific, and the construction of elements of safety measures, safety responsibilities, and safety discipline should be strengthened. In this way, the safety culture is combined with the construction of the safety management system, which increases the binding force of the safety culture.Safety culture guidance. Currently, China’s construction of safety culture is a popular science culture and lacks professional guidance. This paper suggests that in the construction of safety culture, in addition to distinguishing safety culture into safety concept elements, safety discipline elements, safety responsibility elements, and safety measure elements, it should also be differentiated and refined according to different departments, workshops, and positions. The enterprise-level safety culture focuses on the publicity of safety concepts, the department-level safety culture should emphasize responsibility and discipline, and the post-level safety culture should emphasize safety measures and safety responsibilities. The combination of safety culture with job responsibilities makes the control of unsafe acts more informative and constraining.

#### 5.2.2. Safety Culture Construction

Currently, the construction of a safety culture in Chinese businesses is relatively weak and chaotic. The main reasons for this are that (1) the purpose of safety culture construction is not strong, and safety culture lacks binding force and guidance. (2) The means of safety culture construction are single and not combined with the construction of a safety management system. (3) There is no clearly established safety culture element, resulting in a lack of pertinence in the context of safety culture construction. (4) There is a lack of guidance documents for safety culture construction.

It is the view of this paper that the construction of a safety culture should be unified with the construction of a safety management system. In this way, safety policy, safety responsibilities assignment, and safety procedures in the safety management system can be integrated into the elements of safety culture so as to reflect the advantages of safety culture elements in measures, responsibilities, and discipline. At present, there is no unified norm and guiding document on the construction of safety culture elements in the world. This paper suggests that the construction of safety culture elements can be carried out by referring to the safety culture element table (cf. Table 1) proposed by Fu et al. [1,3].

China has only one guiding document, the “*Directives for developing enterprise safety culture*” (AQ/T 9004-2008), in safety culture construction. This guiding document has been used in China for 15 years and its shortcomings are as follows: (1) It stipulates seven basic elements of safety culture construction and the promotion and guarantee mechanism, but it is difficult to operate in actual safety culture construction. (2) There is no clear safety culture element and its corresponding safety content. (3) There is no safety culture construction process.

This paper attempts to strengthen the guidance of safety culture by sorting some international guidance documents on safety culture, as shown in Table 2.

#### 5.2.3. Safety Culture Assessment

A safety culture assessment is an important means to test the construction of a safety culture. In China, the safety culture assessment refers to the “*Assessment standards of enterprise safety culture developing*” (AQ/T 9005-2008). The evaluation indicators proposed by the standards include 11 first-level indicators, 42 second-level indicators, and 144 third-level indicators. At the same time, the safety culture evaluation procedure is also given.

*Safety Culture in Pre-operational Phases of Nuclear Power Plant Projects* (Safety Reports Series No.74) published by the International Atomic Energy Agency [47] provided an explanation for the assessment of safety culture. It differentiates the assessment of safety culture assessment into the pre-assessment and ongoing assessment. Among them, ongoing assessment is further divided into periodic assessment, ongoing monitoring, and continuous improvement. Unfortunately, the project did not propose safety culture evaluation indicators and safety culture evaluation procedures.

Some academics also propose corresponding safety culture assessment indicators based on the characteristics of the evaluation objects, and complete the safety culture assessment by establishing evaluation algorithms [48,49]. The disadvantage of this type of safety culture assessment method is that the evaluation indicators are not comprehensive and lack basis, resulting in inaccurate assessment results.

Based on the aforementioned research, this paper proposes the safety evaluation process as shown in Figure 13 which consists mainly of seven parts. There are currently two main challenges in assessing safety culture.

Lacking guiding documents for safety culture assessment. China’s safety culture assessment refers to the “*Assessment standards of enterprise safety culture developing*” (AQ/T 9005-2008), but it has not been perfected for 15 years and lacks practicality. At present, there is no general guidance document for safety culture assessment in the world.Lacking guiding assessment indicators. The scope of safety culture assessment is relatively wide and involves many industries. However, there is still no unified guiding safety culture assessment index in the world. Each assessment agency sets its own assessment indicators and weights, so the assessment results lack uniformity and accuracy.This paper believes that safety culture assessment can be divided into two categories: safety culture software assessment and safety culture hardware assessment.Safety culture software assessment, which is mainly an assessment of employees’ understanding and recognition of safety culture construction contents, that is, whether employees can put the safety culture construction contents into their minds, which can be achieved by answering questions and systematic questionnaires. In this regard, Fu Gui designed a safety culture assessment question bank based on the safety culture element table (cf. Table 1) and developed a safety culture analysis program [43], which can be referred to for safety culture assessment.Safety culture hardware assessment, which mainly evaluates whether the organizational structure of safety culture construction is complete, whether the construction measures are appropriate, and whether the safety culture carrier is effective. This evaluation is more complicated.

Overall, there is an urgent and important need for the development of a guiding safety culture assessment index and process at an international level. It can effectively guide different countries, regions, and businesses to conduct safety culture assessment and construction and play a significant role in controlling unsafe acts.

### 5.3. Safety Management System

Based on the simulation results, the influence of safety management system on unsafe acts presents two laws. In new coalmines, the order of impact is as follows: safety policy > safety management organizational structure > safety management procedures. Currently, safety policy has a strong impact on unsafe acts, while the gap between safety management organizational structure and safety management procedures is small. In production coalmines, the ranking is safety management organizational structure > safety management procedures > safety policy. There is little difference in the impact between the three.

Research has shown that the simulation results mentioned above are consistent with actual safety management laws. Safety policy refers to the general requirements of the government, industry, and companies for safety production, and it is the direction of safety production. In new coalmines, the safety policy is like a safety concept that directly affects the safety attitude of the employee, so it first reflects the effect in the control of unsafe acts. Safety management organizational structure involves safety organization setup and staffing, while safety management procedures involve numerous and complex procedures, so their effect on unsafe acts is not as significant as a safety policy. In production coalmines, the safety management organizational structure already has an accurate mechanism setup, responsibility assignment, and personnel allocation, and the safety management procedures are perfect and run smoothly. Therefore, both have a strong influence on unsafe acts (cf. Figure 14).

In modern safety management, companies pay attention to clear safety policies, perfect safety management organizational structure, and programmed safety management processes. These three are interdependent in the process of safety management and jointly affect the safety behavior of employees. Therefore, in the production coalmine, the difference in the impact of the three on unsafe acts is small. Overall, a company’s ability to manage safety can be seen from the perfection and functioning of the company’s safety management system.

Through the study of coal and gas outburst accidents, Xie et al. [7] found that the problems of the safety management system are mainly reflected as follows:Some companies do not follow the national and corporate safety policy, which is illegal production.In the organizational structure of safety management, the mechanism setup is not perfect, the responsibility assignment is not clear, and the personnel allocation is insufficient (especially technical personnel and department heads).The enterprise safety management program is missing or not running smoothly.

How to guide the construction of an enterprise safety management system plays an important role in enterprise safety production. *Occupational health and safety management systems—Requirements with guidance for use* (GB/T 45001-2020 /ISO45001:2018) is the current official guidance document for the construction of a Chinese enterprise safety management system. However, its content is too broad and lacks industry safety content, so it is not effective in practical applications. This paper suggests that the Ministry of Emergency Management of China should combine the characteristics of industry safety management and propose guidance documents for the construction of a safety management system by industry. In addition, it is necessary to organize industry experts to conduct training on the construction of a safety management system and inspect its construction and operation. This is the most effective way to build a good enterprise safety management system. Unfortunately, China still lacks enterprise safety management systems and construction evaluation methods.

### 5.4. Individual Safety Ability

Individual safety ability is the direct reflection of the safety quality of employees, which is affected by the enterprise’s safety culture and safety management system and directly affects individual safety acts. Individual safety ability includes many factors. From accident analysis, the most direct factors are safety knowledge, safety awareness, safety habit, and safety psychology [1,3,7,12]. From the simulation results, whether it was a new coalmine or a production coalmine, the degree of influence on the safety ability was safety knowledge > safety psychology = safety habits > safety awareness. However, the effect on the safety ability level was minimal, and the approximation could be considered consistent. Fu et al. [3] gave the law of mutual influence between them. According to the cause of the accident, the four factors can be further excavated and more finely divided, which are conducive to the construction of safety ability and the management of unsafe acts (cf. Figure 15).

#### 5.4.1. Safety Knowledge

Safety knowledge is professional knowledge closely related to safety production. Through the accidents analysis, Xie et al. [7] found that safety knowledge can be roughly divided into two categories: safety management knowledge and professional skills knowledge (cf. Figure 16).

Safety management knowledge is mainly the knowledge of enterprise safety systems, including safety culture knowledge, safety management organizational structure knowledge, safety procedures knowledge, safety management responsibility system knowledge, etc.Professional skills knowledge is mainly the knowledge of professional skills in safety operations, including standardized job operation knowledge, professional equipment operation knowledge, emergency response knowledge, etc.

For the acquisition of safety knowledge, safety training is an important way to go. This paper recommends that safety training should do the following:Safety knowledge should be comprehensive. The enterprise’s safety department should systematically sort out the reasons for accidents and compile an accident reasons database. At the same time, it is necessary to combine knowledge of safety procedures and policies. This allows employees to gain comprehensive knowledge of safety management and professional skills.For safety management knowledge training, it is encouraged to adopt logical and clear methods such as flowcharts, which can make it easier for employees to understand the operating mode of the company’s safety management system.For professional skills knowledge training, it is necessary to clarify specific terms, operation methods, operation procedures, etc., so as to accurately master professional skills.This paper proposes that in safety knowledge training, different categories of employees should conduct safety training based on their work priorities. Senior leadership should focus on training national and industrial safety policies, safety management decisions, and so on. Middle management should focus on cultivating industry standards, norms and actions, etc. Frontline staff should focus on cultivating standardized post-operations skills, equipment operations skills, and emergency response skills.Conventional methods of safety training (such as lectures) rely primarily on rapid knowledge transfer. It has a relatively boring and monotonous classroom format and lacks flexibility and activity. Consequently, the adoption of new modes of instruction such as video instruction, VR experience instruction, and on-site instruction will be closer to reality, and the safety training effect will be better.Paying attention to assessing the effect of employee education. Employees will find it easier to accept a more visual, modular, and convenient assessment model than traditional assessment methods.

#### 5.4.2. Safety Awareness

Safety awareness is the ability to sense, detect, and resolve hazards in a timely manner. This paper found that safety awareness can be briefly classified into three categories (cf. Figure 17).

Safety regulation awareness: whether employees are aware of whether their behavior violates safety standards, safety regulations, safety procedures, etc.Safety risk awareness: whether employees are aware that their acts will lead to risks or accidents.Safety responsibility awareness: whether employees are aware that their behavior violates their job responsibilities, that is, whether they have implemented a safety responsibility system.

This paper presents methods to improve safety awareness (cf. Figure 17). Among them, the most important method is safety training. During safety training, careful analysis of specific categories of employees’ dangerous awareness is required when each dangerous act occurs. The safety department should triage and summarize frequently occurring unsafe awareness and unsafe acts and then take safety measures and focus on training (cf. Figure 17). Based on accident analysis [7], poor safety regulation awareness is the most frequent, accounting for about 60%, which needs to be paid attention to.

#### 5.4.3. Safety Habit

Safety habits are frequently repeated acts in daily work, which are affected by organizational factors (safety culture and safety management system) and individual safety knowledge, safety awareness, safety psychology, and other aspects. Unsafe acts are direct accident reasons that can be divided into one-time violations and habitual violations. The most common unsafe behavior among safety habits is habitual violations (cf. Figure 18).

Habitual violations are essentially high-frequency unsafe acts, which are mainly caused by inadequate safety training and safety supervision. Habitual violations are “contagious”; its spread will seriously reduce employees’ safety psychology and safety attitude and destroy the enterprise safety atmosphere. In safety management, the “enterprise-workshop-team” should promptly discover and sort out employees’ violations and establish a habitual violation database. For habitual violations, the correction should be done “demonstration—correction—observation—consolidation” (such as Behavior-Based Safety), and it is necessary to strictly prevent its existence and spread.

#### 5.4.4. Safety Psychology

Safety psychology is the psychological state of employees during their daily work and represents their attitude toward safety. Employees’ unsafe psychology will directly lead to unsafe acts. There are many kinds of safety psychology, which can be roughly divided into four types (cf. Figure 19). Viewing from accident analysis [7], fluke psychology has the highest frequency (about 40%) and needs to be paid attention to.

The most important way to overcome unsafe psychology is safety training, especially a training method that is closer to the accident and will bring about a more real “accident stimulation” feeling. Safety managers should analyze the psychological status of personnel who send unsafe acts as part of daily safety management, as well as establish an unsafe psychological database. When employees are found to be in an unsafe psychological state at work, the safety manager should intervene and correct them in time.

Psychological assessments are used in many industries [50,51] with many successful outcomes. Establishing a set of methods to assess the employees’ safety psychology will have a significant effect on controlling unsafe acts and reducing accidents, but there are relatively few studies in this area [52,53]. This paper suggests that it is very necessary to establish a set of safety psychological evaluation mechanisms based on the full study of “safety knowledge-safety psychology-safety awareness-safety habits-safety acts”.

#### 5.4.5. Influence Mechanism of Individual Safety Ability on Unsafe Acts

Revealing the causes of unsafe acts from the level of the mechanism is useful for understanding the entire process of unsafe acts. Currently, the behavior mechanism of unsafe acts is still unclear [54,55]. This paper combines the above analysis and simulation results to propose unsafe acts’ mechanisms at the level of individual safety capability (cf. Figure 20).

Undoubtedly, safety knowledge is the most important way to improve individual safety ability. Safety knowledge can be obtained through organizational-level safety training or through individual self-learning. For employees, safety knowledge is like a “stimulus signal”. When employees receive the “stimulus”, they do not have an immediate “safety response”. The stimulus must reach the individual’s “threshold value” to produce qualitative changes, which, in turn, lead to changes in the individual’s safety psychology. Of course, the “threshold value” of the stimulus varies with the employee’s position, type of work, education, and so on. The ways of “stimulating signals” of safety knowledge are different, and the consequences are also different. Under normal circumstances, more specific, practical, and realistic safety training methods are more easily accepted by employees. When the stimulus “threshold value” is exceeded, the employee’s safety ability will undergo qualitative changes. This is manifested in the improvement of safety psychology and safety awareness, the reduction of habitual violations, and the increase of safety operations, which in turn leads to the formation of stable safety acts by employees.

Through the analysis, it can be seen that the increase in individual safety ability focuses on the content, method, and intensity of “stimulation”. Therefore, during safety training, the training content should be specific and easy to understand and remember. Training should be targeted, that is, training should be focused on different groups of people. At the same time, the training method should pay attention to diversity. Usually, visual, perceptual, and on-site training methods will produce better “stimulus” effects.

Assessment of safety ability is of great assistance in measuring employees’ safety qualities [56]. Assessment of employees’ safety knowledge should become a mandatory course for safety education. Observing and correcting habitual violations has always been a major task and day-to-day job of the corporate safety supervisory department. One of the most easily overlooked aspects of daily safety management is the assessment of safety awareness and safety psychology, which are important ways to reduce habitual violations. In this paper, we believe that in the safety management aspect, doing a good job of building and evaluating safety awareness and safety psychology is worthy of careful study.

## 6. Conclusions

In terms of systems thinking, the study of corporate safety system management and control laws for employees’ unsafe acts plays a significant role in accident prevention. By considering coal and gas outburst accidents as the research focus, this study adopts a system dynamics method to investigate the influence of corporate safety culture, safety management system, and individual safety ability on individual safety acts. The main conclusions of this study are as follows:For new coalmines and low safety level production coalmines, this study proposes building an enterprise safety culture, safety management system, and individual safety ability according to the maximum safety ability of the enterprise. For an enterprise with a high level of safety, this study proposes certain optimizations and adjustments of the safety construction measures to be performed. However, it is not recommended to perform large-scale safety construction to avoid disrupting the high-quality safety management methods that employees are accustomed to, which reduces the safety level.In new coalmines, the influence of the safety culture, safety management system, and safety ability on the safety acts was similar. In production coalmines, the order of influence on the safety acts was as follows: safety management system > safety ability > safety culture. The difference is most evident in months ten to eighteen. The greater the level of safety and the safety construction standard of the enterprise, the greater the difference.In the construction of safety culture, the order of influence was as follows: safety measure elements > safety responsibility elements = safety discipline elements > safety concept elements. From the sixth month, it shows the difference in influence and reaches the maximum value from the twelfth to the fourteenth month. In the construction of the safety management system, the degree of influence in new coalmines was as follows: safety policy > safety management organization structure > safety management procedures. Among them, especially in the first 18 months, the impact of the safety policy is most apparent. However, in the production mine, the degree of influence was as follows: safety management organization structure > safety management procedures > safety policy, but the difference is very weak. In the construction of safety ability, the degree of influence was as follows: safety knowledge > safety psychology = safety habits > safety awareness, but the difference in the impact was small.Based on the simulation results and the actual safety management, this paper analyzes the occurrence mechanism of unsafe acts. At the same time, this paper simultaneously sets out the measures and methods of safety culture construction and evaluation, safety management system construction, and safety ability construction so as to control the occurrence of unsafe acts. Based on the mechanism of unsafe acts, the control of unsafe acts should be implemented jointly from the safety culture, safety management system, and safety ability.

## Figures and Tables

**Figure 2 ijerph-20-04733-f002:**
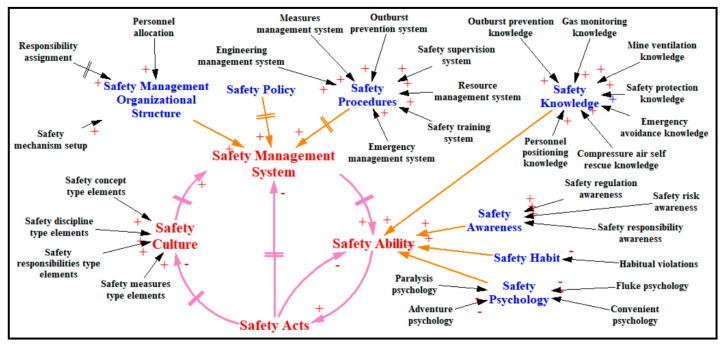
Causal relationship for unsafe acts in the coal and gas outburst accidents.

**Figure 3 ijerph-20-04733-f003:**
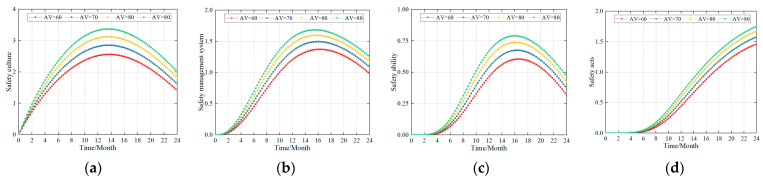
Simulation images of the safety culture, safety management system, safety ability, and safety acts in the new coalmine. (**a**) Safety culture; (**b**) safety management system; (**c**) safety ability; and (**d**) safety acts.

**Figure 4 ijerph-20-04733-f004:**
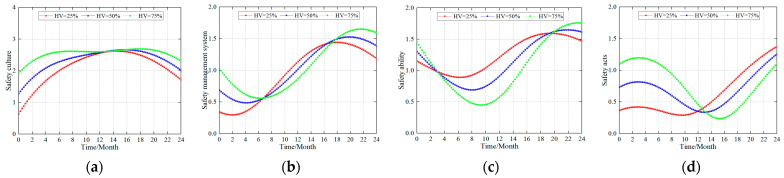
Simulation images of the safety culture, safety management system, safety ability, and safety acts in the production coalmine. (**a**) Safety culture; (**b**) safety management system; (**c**) safety ability; and (**d**) safety acts.

**Figure 5 ijerph-20-04733-f005:**
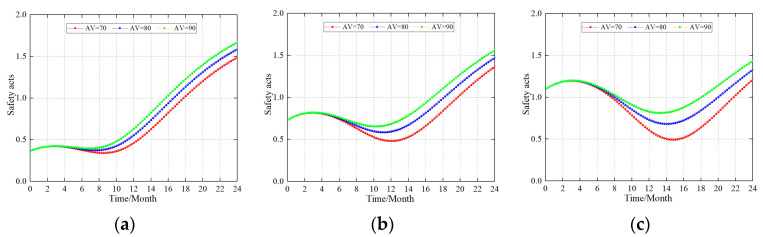
Changes in the safety acts for the production mine with different safety building abilities. (**a**) HV = 25%; (**b**) HV = 50%; and (**c**) HV = 75%.

**Figure 6 ijerph-20-04733-f006:**
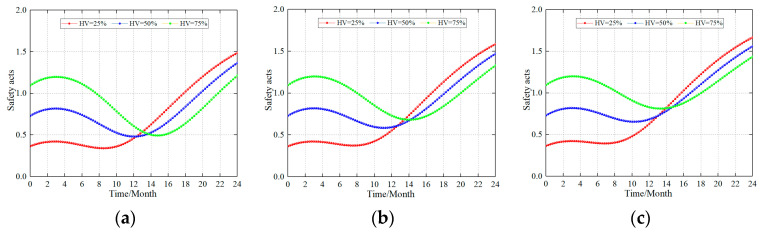
Changes in the safety acts of the production mine under different safety levels. (**a**) AV = 70; (**b**) AV = 80; and (**c**) AV = 90.

**Figure 7 ijerph-20-04733-f007:**
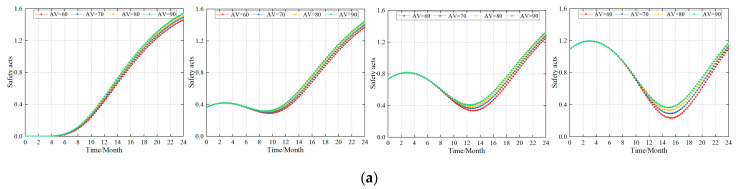
Simulation images for the impact of the safety culture on the safety acts. (**a**) Influence of the safety culture on the safety acts (HV = 0%, HV = 25%, HV = 50%, and HV = 75%); (**b**) influence of the safety management system on the safety acts (HV = 0%, HV = 25%, HV = 50%, and HV = 75%); and (**c**) influence of the safety ability on the safety acts (HV = 0%, HV = 25%, HV = 50%, and HV = 75%).

**Figure 8 ijerph-20-04733-f008:**
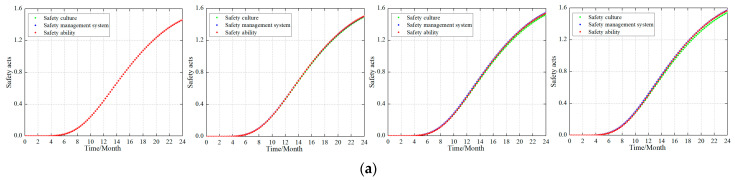
Impact of the different safety cultures, safety management systems, and safety ability building standards on the safety acts. (**a**) HV = 0%, AV = 60, 70, 80, and 90; (**b**) HV = 25%, AV = 60, 70, 80, and 90; (**c**) HV = 50%, AV = 60, 70, 80, and 90; and (**d**) HV = 75%, AV = 60, 70, 80, and 90.

**Figure 10 ijerph-20-04733-f010:**
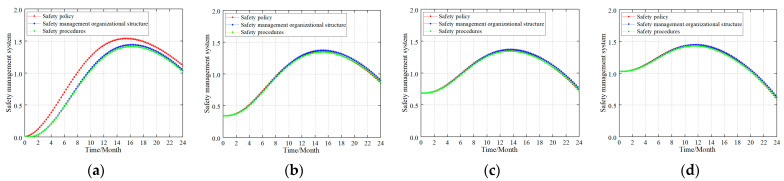
Effect of the different safety management system factors on the safety level of the safety management system. (**a**) HV = 0%; (**b**) HV = 25%; (**c**) HV = 50%; and (**d**) HV = 75%.

**Figure 11 ijerph-20-04733-f011:**
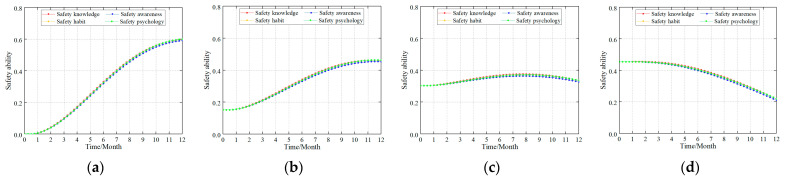
Effect of the different safety ability factors on the safety ability level. (**a**) HV = 0%; (**b**) HV = 25%; (**c**) HV = 50%; and (**d**) HV = 75%.

**Figure 12 ijerph-20-04733-f012:**
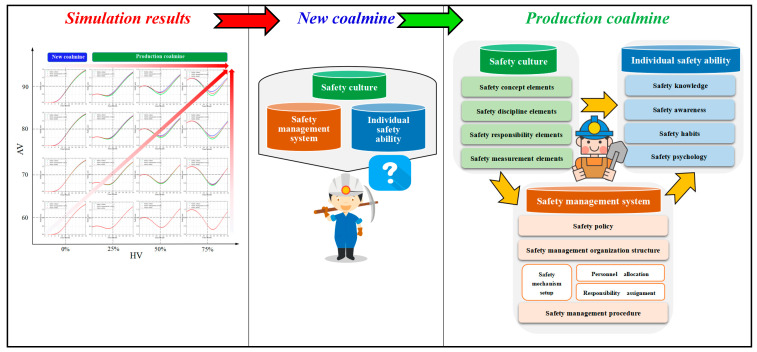
The behavioral mechanism that affects unsafe acts at the enterprise safety system level.

**Figure 13 ijerph-20-04733-f013:**
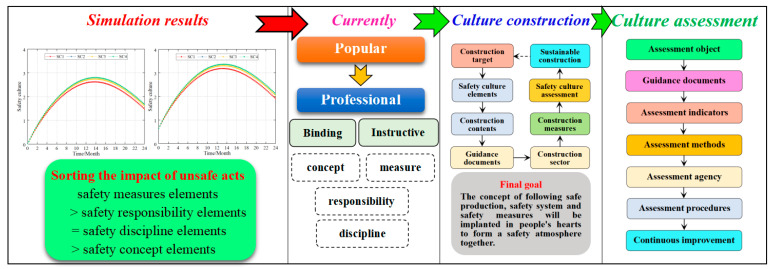
Current situation, problems, and methods in safety culture construction and assessment.

**Figure 14 ijerph-20-04733-f014:**
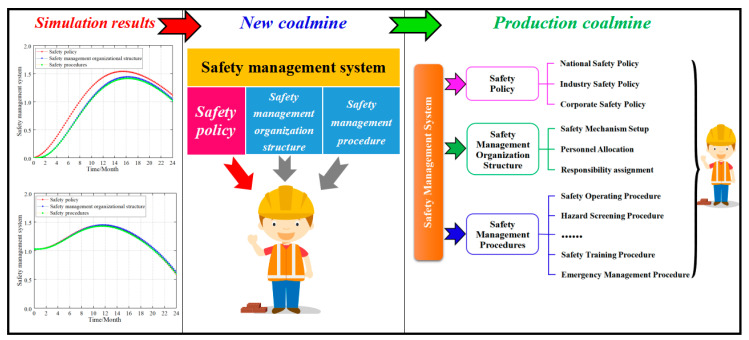
Influence law of safety management system on unsafe acts and its evolution characteristics.

**Figure 15 ijerph-20-04733-f015:**
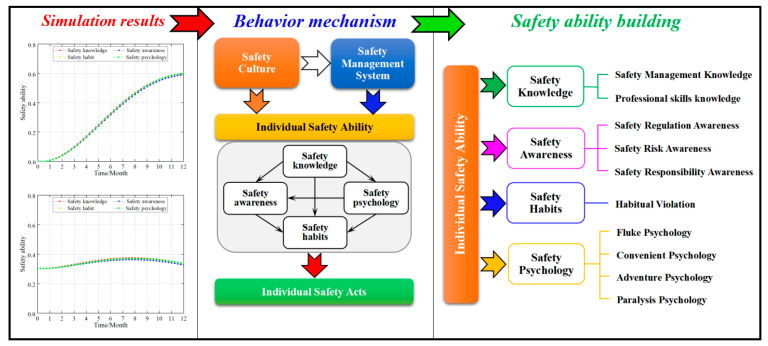
Influence mechanism and construction content of individual safety ability.

**Figure 16 ijerph-20-04733-f016:**
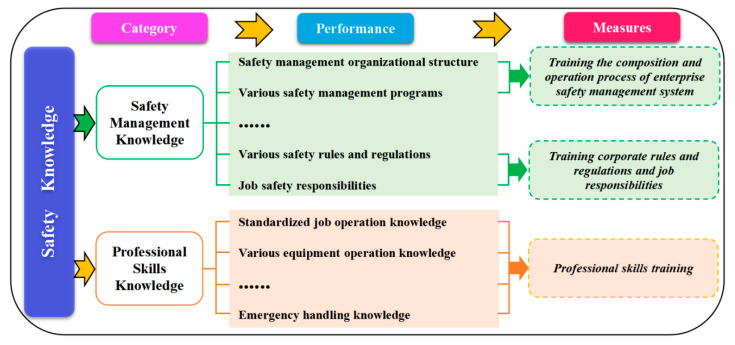
Classification and acquisition of safety knowledge.

**Figure 17 ijerph-20-04733-f017:**
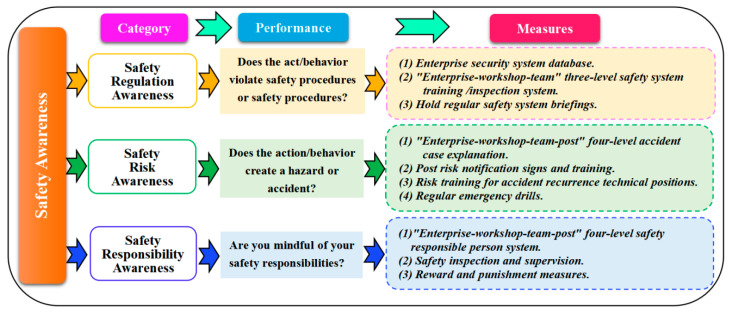
Classification and acquisition of safety awareness.

**Figure 18 ijerph-20-04733-f018:**
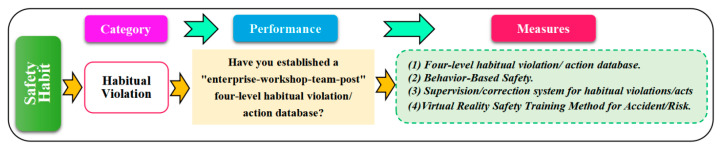
Classification and acquisition of safety habits.

**Figure 19 ijerph-20-04733-f019:**
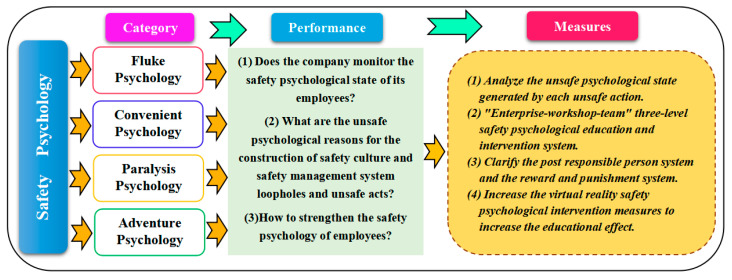
Classification and acquisition of safety psychology.

**Figure 20 ijerph-20-04733-f020:**
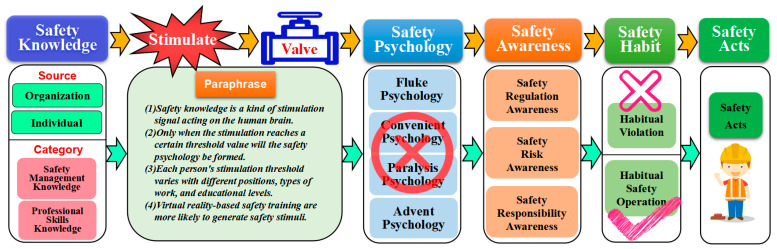
Induction mechanism of individual safety ability to safety acts.

**Table 1 ijerph-20-04733-t001:** Safety culture elements table in the 24Model [1,3,7,12].

No.	Safety Culture Element Name	No.	Safety Culture Element Name
SCE1	Safety Importance	SCE17	Safety Meeting
SCE2	Beliefs of Zero Accidents	SCE18	Formation of Safety Rules
SCE3	Trade-off between Safety and Profit	SCE19	Enforcement of Safety Rules
SCE4	Safety and Management Integration	SCE20	Injury and Incident Investigation
SCE5	Safety Awareness	SCE21	Workplace Audits, Inspections
SCE6	Primary Responsibility for Workplace Safety	SCE22	Modified Duty and Return-to-Work Systems
SCE7	Safety Investment	SCE23	Off-the-Job Safety
SCE8	Safety Regulations	SCE24	Recognition for Safety Performance
SCE9	Safety Values	SCE25	Facility Satisfaction
SCE10	Safety Responsibility of Mangers	SCE26	Measuring and Bench marking Safety Performance
SCE11	Role of Safety Department	SCE27	Hiring for Safety Attitude
SCE12	Individual Involvement in Safety	SCE28	Safety of Contractors and Subsidiaries
SCE13	Demand of Safety Training	SCE29	Safety Organization
SCE14	Responsibilities for Safety in Every Department	SCE30	Safety Department
SCE15	Involvement in Community Safety	SCE31	Overall Expectation for Safety
SCE16	Role of Safety Management System	SCE32	Emergency Capability
	**SCE1–SCE3: Safety concept elements**		**SCE9–SCE16: Safety responsibility elements**
	**SCE4–SCE8: Safety discipline elements**		**SCE17–SCE32: Safety measurement elements**

**Table 2 ijerph-20-04733-t002:** Safety culture construction directives.

No	Year	Publisher	Directives	Main Contribution
1	1991	IAEA	Safety Culture (No.75-INSAG-4)	(1) Definition of safety culture (2) Strategy for the construction of nuclear safety culture
2	1994	IAEA	TECDOC 743: ASCOT Guidelines―Guidelines for Organizational Self-Assessment of Safety Culture and for Reviews by the Assessment of Safety Culture in Organizations Team	(1) Structure of the ASCOT guidelines (Government and (2) its organizations, Corporate level, Plant level, Support organizations) (3) ASCOT services
3	1997	IAEA	Safety Reports Series No.1: Examples of Safety Culture Practices	(1) Examples of safety culture practices (2) Effecting safety culture improvement
4	1998	IAEA	Safety Reports Series No.11: Developing Safety Culture in Nuclear Activities—Practical Suggestions to Assist Progress	(1) Described practices valuable in establishing and maintaining a sound safety culture in a number of countries. (2) Provided a reference for groups such as regulators who have an interest in developing, improving, and evaluating safety culture training and individuals engaged in nuclear activities.
5	2002	IAEA	Key Practical Issues in Strengthening Safety Culture (INSAG 15)	(1) This report describes the essential practical issues to be considered by organizations aiming to strengthen safety culture.
6	2002	IAEA	(1) Self-Assessment of Safety Culture in Nuclear Installations Highlights and Good Practices (TECDOC 1321) (2) Safety Culture in Nuclear Installations: Guidance for Use in the enhancement of Safety Culture (TECDOC 1329)	(1) What is safety culture (2) Stage of development of safety culture (3) Self-assessment of safety culture (4) Practice of development of safety culture (5) Safety culture indicators (6) Change the safety culture
7	2003	ILO	91st International Labour Conference	(1) Proposed the concept of “National Preventive Safety and Health Safety”.
8	2006	ILO	(1) Promotional Framework for Occupational Safety and Health Convention (C187, 2006) (2) Promotional Framework for Occupational Safety and Health Recommendation(R197, 2006)	
9	2008	MEMPRC	Directives for developing enterprise safety culture (AQ/T 9004-2008)	(1) The *Directives* stipulate seven basic elements of safety culture construction and the promotion and guarantee mechanism.
10	2012	IAEA	Safety Reports Series No.74: Safety Culture in Pre-operational Phases of Nuclear Power Plant Projects	(1) Safety culture relevance to pre-operational phases. (2) Understanding nuclear safety and safety culture (3) Management system processes to support the safety culture (4) Cultural assessment and continuous improvement
11	2016	IAEA	Safety Reports Series No.83: Performing Safety Culture Self-Assessments	(1) This publication provides practical guidance on how to conduct a safety culture self-assessment.
**(1) International Atomic Energy Agency = IAEA, (2) International Labour Organization = ILO,** **(3) Ministry of Emergency Management of the People’s Republic of China = MEMPRC**				

## Data Availability

Not applicable.

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
