# Peer review of "Investigation of Unsafe Acts Influence Law Based on System Dynamics Simulation: Thoughts on Behavior Mechanism and Safety Control"

_ijerph, 2023, doi:10.3390/ijerph20064733_

Round 1
Reviewer 1 Report
The manuscript adopted a system dynamics method to study the influence of enterprise safety culture, safety management system and individual safety ability on the individual safety acts. The paper is interesting, with many simulation images and concepts. However several issues should be solved before it can be accepted.
1. English text needs to be improved and further editing is necessary.
2. The conclusions and abstract are not quantitative. The authors should explore more data from Figure 3 to Figure11.
3. Page 5. The equations are very non-standard. Authors need to pay attention to superscript and subscript. And the difference between * and × needs to be unified.
4.How to verify the effectiveness of your system dynamics model?
Reviewer 2 Report
Despite being a very interesting theme, it loses interest by focusing on a specific area and not using, for example, a comparison of the different existing references.
The structure of the article is a little confusing in terms of content because there is no distinction of information between the different chapters. It is written not focused on a scientific paper but as a thesis.
You can find the suggestions along the document.
I would suggest a revision of the whole document before re-submission.

Round 2
Reviewer 2 Report
The paper presents significant improvements in its structure and presentation of content. I leave only for their consideration improvements in the points indicated in the manuscript.
